# A Classification–Based Perspective on GAN Distributions

## Abstract

A fundamental, and still largely unanswered, question in the context of Generative Adversarial Networks (GANs) is whether GANs are actually able to capture the key characteristics of the datasets they are trained on. The current approaches to examining this issue require significant human supervision, such as visual inspection of sampled images, and often offer only fairly limited scalability. In this paper, we propose new techniques that employ classification–based perspective to evaluate synthetic GAN distributions and their capability to accurately reflect the essential properties of the training data. These techniques require only minimal human supervision and can easily be scaled and adapted to evaluate a variety of state-of-the-art GANs on large, popular datasets. They also indicate that GANs have significant problems in reproducing the more distributional properties of the training dataset. In particular, the diversity of such synthetic data is orders of magnitude smaller than that of the original data.

## 1 Introduction

Generative Adversarial Networks (GANs) (Goodfellow et al., 2014) have garnered a significant amount of attention due to their ability to learn generative models of multiple natural image datasets (Radford et al., 2015; Denton et al., 2015; Zhang et al., 2016; Zhu et al., 2017). Since their conception, a fundamental question regarding GANs is to what extent they truly learn the underlying data distribution. This is a key issue for multiple reasons. From a scientific perspective, understanding the capabilities of common GANs can shed light on what precisely the adversarial training setup allows the GAN to learn. From an engineering standpoint, it is important to grasp the power and limitations of the GAN framework when applying it in concrete applications. Due to the broad potential applicability of GANs, researchers have investigated this question in a variety of ways.

When we evaluate the quality of a GAN, an obvious first check is to establish that the generated samples lie in the support of the true distribution. In the case of images, this corresponds to checking if the generated samples look realistic. Indeed, visual inspection of generated images is currently the most common way of assessing the quality of a given GAN. Individual humans can performs this task quickly and reliably, and various GANs have achieved impressive results for generating realistic-looking images of faces and indoor scenes (Salimans et al., 2016; Denton et al., 2015).

Once we have established that GANs produce realistic-looking images, the next concern is that the GAN might simply be memorizing the training dataset. While this hypothesis cannot be ruled out entirely, there is evidence that GANs perform at least some non-trivial modeling of the unknown distribution. Previous studies show that interpolations in the latent space of the generator produce novel and meaningful image variations (Radford et al., 2015), and that there is a clear disparity between generated samples and their nearest neighbors in the true dataset (Arora & Zhang, 2017).

Taken together, these results provide evidence that GANs could constitute successful distribution learning algorithms, which motivates studying their distributions in more detail. The direct approach is to compare the probability density assigned by the generator with estimates of the true distribution (Wu et al., 2016). However, in the context of GANs and high-dimensional image distributions, this is complicated by two factors. First, GANs do not naturally provide probability estimates for their samples. Second, estimating the probability density of the true distribution is a challenging problem itself (the adversarial training framework specifically avoids this issue). Hence prior work has only investigated the probability density of GANs on simple datasets such as MNIST (Wu et al., 2016).

Since reliably computing probability densities in high dimensions is challenging, we can instead study the behavior of GANs in low-dimensional problems such as two-dimensional Gaussian mixtures. Here, a common failure of GANs is *mode collapse*, wherein the generator assigns a disproportionately large mass to a subset of modes from the true distribution (Goodfellow, 2016). This raises concerns about a lack of diversity in the synthetic GAN distributions, and recent work shows that the learned distributions of two common GANs indeed have (moderately) low support size for the CelebA dataset (Arora & Zhang, 2017). However, the approach of Arora & Zhang (2017) heavily relies on a human annotator in order to identify duplicates. Hence it does not easily scale to comparing many variants of GANs or asking more fine-grained questions than collision statistics. Overall, our understanding of synthetic GAN distributions remains blurry, largely due to the lack of versatile tools for a quantitative evaluation of GANs in realistic settings. The focus of this work is precisly to address this question:

> *Can we develop principled and quantitative approaches to study synthetic GAN distributions?*

To this end, we propose two new evaluation techniques for synthetic GAN distributions. Our methods are inspired by the idea of comparing moments of distributions, which is at the heart of many methods in classical statistics. Although simple moments of high-dimensional distributions are often not semantically meaningful, we can extend this idea to distributions of realistic images by leveraging image statistics identified using convolutional neural networks. In particular, we train image classifiers in order to construct test functions corresponding to semantically meaningful properties of the distributions. An important feature of our approach is that it requires only light human supervision and can easily be scaled to evaluating many GANs and large synthetic datasets.

Using our new evaluation techniques, we study five state-of-the-art GANs on the CelebA and LSUN datasets, arguably the two most common testbeds for advanced GANs. We find that most of the GANs significantly distort the relative frequency of even basic image attributes, such as the hair style of a person or the type of room in an indoor scene. This clearly indicates a mismatch between the true and synthetic distributions. Moreover, we conduct experiments to explore the diversity of GAN distributions. We use synthetic GAN data to train image classifiers and find that these have significantly lower accuracy than classifiers trained on the true data set. This points towards a lack of diversity in the GAN data, and again towards a discrepancy between the true and synthetic distributions. In fact, our additional examinations show that the diversity in GANs is only comparable to a subset of the true data that is $100\times$ smaller.

## 2  UNDERSTANDING GANS THROUGH THE LENS OF CLASSIFICATION

When comparing two distributions, a common first test is to compute low-order moments such as the mean and the variance. If the distributions are simple enough, these quantities provide a good understanding for how similar they are. Moreover, low-order moments have a precise definition and are usually quick to compute. On the other hand, low-order moments can also be misleading for more complicated, high-dimensional distributions. As a concrete example, consider a generative model of digits (such as MNIST). If a generator produces digits that are shifted by a significant amount yet otherwise perfect, we will probably still consider this as a good approximation of the true distribution. However, the expectation (mean moment) of the generator distribution can be very different from the expectation of the true data distribution. This raises the question of what other properties of high-dimensional image distributions are easy to test yet semantically meaningful.

In the next two subsections, we describe two concrete approaches to evaluate synthetic GAN data that are easy to compute yet capture relevant information about the distribution. The common theme is that we employ convolutional neural networks in order to capture properties of the distributions that are hard to describe in a mathematically precise way, but usually well-defined for a human (e.g., what fraction of the images shows a smiling person?). Automating the process of annotating images with such high-level information will allow us to study various aspects of synthetic GAN data.

### 2.1  QUANTIFYING MODE COLLAPSE

Mode collapse refers to the tendency of the generator to concentrate a large probability mass on a few modes of the true distribution. While there is ample evidence for the presence of mode-collapse in GANs (Goodfellow, 2016; Arora & Zhang, 2017; Metz et al., 2016), elegant visualizations of this phenomena are somewhat restricted to toy problems on low-dimensional distributions (Goodfellow,

2016; Metz et al., 2016). For image datasets, it is common to rely on human annotators and derived heuristics (see Section 2.3). While these methods have their merits, they are restrictive both in the scale and granularity of testing. Here we propose a classification-based tool to assess how good GANs are at assigning the right mass across broad concepts/modes. To do so, we use a trained classifier as an expert "annotator" that labels important features in synthetic data, and then analyze the resulting distribution. Specifically, our goal is to investigate if a GAN trained on a well-balanced dataset (i.e., contains equal number of samples from each class) can learn to reproduce this balanced structure. Let $D = (X, Y) = \{(x_i, y_i)\}_{i=1}^N$ represent a dataset of size $N$ with $C$ classes, where $(x_i, y_i)$ denote an image-label pair drawn from true data. If the dataset $D$ is balanced, it contains $N/C$ images per class. The procedure for computing class distribution in synthetic data is:

1. Train an annotator (a multi-class classifier) using the dataset $D$.
2. Train an unconditional GAN on the images $X$ from dataset $D$, without using class labels.
3. Create a synthetic dataset by sampling $N$ images from a GAN and labeling them using the annotator from Step 1.

The annotated data generated via the above procedure can provide insight into the GAN's class distribution at the scale of the entire dataset. Moreover, we can vary the granularity of mode analysis by choosing richer classification tasks, i.e., more challenging classes or a larger number of them. In Section 3.3, we use this technique to visualize mode collapse in several state-of-the-art GANs on the CelebA and LSUN datasets. All the studied GANs show significant mode collapse and the effect becomes more pronounced when the granularity of the annotator is increased (larger number of classes). We also investigate the temporal aspect of the GAN setup and find that the dominant mode varies widely over the course of training. Our approach also enables us to benchmark and compare GANs on different datasets based on the extent of mode collapse in the learned distributions.

## 2.2 MEASURING DIVERSITY

Our above method for inspecting distribution of modes in synthetic data provides a coarse look at the statistics of the underlying distribution. While the resulting quantities are semantically meaningful, they capture only simple notions of diversity. To get a more holistic view on the sample diversity in the synthetic distribution, we now describe a second classification-based approach for evaluating GAN distributions. The main question that motivates it is: Can GANs recover the key aspects of real data to enable training a good classifier? We believe that this is an interesting measure of sample diversity for two reasons. First, classification of high-dimensional image data is a challenging problem, so a good training dataset will require a sufficiently diverse sample from the distribution. Second, augmenting data for classification problems is one of the proposed use cases of GANs (e.g., see the recent work of Shrivastava et al. (2017)).

If GANs are truly able to capture the quality and diversity of the underlying data distribution, we expect almost no gap between classifiers trained on true data and synthetic data from a GAN. A generic method to produce data from GANs for classification is to train separate GANs for each class in the dataset $D$.[1] Samples from these class-wise GANs can then be pooled together to get a labeled synthetic dataset. Note that the labels are trivially determined based on the class modeled by the particular GAN from which a sample is drawn. We perform the following steps to assess the classification performance of synthetic data vs. true data:

1. Train a classifier on the true data $D$ (from Section 2.1) as a benchmark for comparison.
2. Train $C$ separate unconditional GANs, one per class in dataset D.
3. Generate a balanced synthetic labeled dataset of size N by consolidating an equal number of samples drawn from each of these $C$ GANs. The labels obtained by aggregating samples from per-class GANs are designated as "default" labels for the synthetic dataset. Note that by design, both true and synthetic datasets have $N$ samples, with $N/C$ examples per class.
4. Use synthetic labeled data from Step 3 to train classifier with the same architecture as Step 1.

Comparing the classifiers from Steps 1 and 4 can then shed light on the disparity between the two distributions. Radford et al. (2015) conducted an experiment similar to Step 2 on the MNIST dataset using a *conditional* GAN. They found that samples from their DCGAN performed comparably to true data on nearest neighbor classification. We obtained similar good results on MNIST, which

---

[1]We tried the alternate approach of using class-conditional GANs to get labeled datasets. This method did not yield good samples since most common GANs have not been designed with a conditional structure in place.

could be due to the efficacy of GANs in learning the MNIST distribution or due to the ease of getting good accuracy on MNIST even with a small training set (Rolnick et al., 2017). To clarify this question, we restrict our analysis to more complex datasets, specifically CelebA and LSUN.

We evaluate the two following properties in our classification task:

(i) How well can the GANs recover nuances of the decision boundary, which is reflected by how easily the classifier can fit the training data?

(ii) How does the diversity of synthetic data compare to that of true data when measured by classification accuracy on a hold-out set of true data?

We observe that all the studied GANs have very low diversity in this metric. In particular, the accuracy achieved by a classifier trained on GAN data is comparable only to the accuracy of a classifier trained on a $100\times$ (or more) subsampled version of the true dataset. Even if we draw more samples from the GANs to produce a training set several times larger than the true dataset, there is no improvement in performance. Looking at the classification accuracy gives us a way to compare different models on a potential downstream application of GANs. Interestingly, we find that visual quality of samples does not necessarily correlate with good classification performance.

## 2.3 RELATED WORK

In GAN literature, it is common to investigate performance using metrics that involve human supervision. Arora & Zhang (2017) proposed a measure based on manually counting duplicates in GAN samples as a heuristic for the support or diversity of the learned distribution. In Wu et al. (2016), manual classification of a small sample (100 images) of GAN generated MNIST images is used as a test for the GAN is missing certain modes. Such annotator-based metrics have clear advantages in identifying relevant failure-modes of synthetic samples, which explains why visual inspection (eyeballing) is still the most popular approach to assess GAN samples.

There have also been various attempts to build good metrics for GANs that are not based on manual heuristics. Parzen window estimation can be used to approximate the log-likelihood of the distribution, though it is known to work poorly for high-dimensional data (Theis et al., 2016). Wu et al. (2016) develop a method to get a better estimate for log-likelihood using annealed importance sampling. Salimans et al. (2016) propose a metric known as Inception Score, where the entropy in the labels predicted by a pre-trained Inception network is used to assess the diversity in GAN samples.

## 3 EXPERIMENTS

In the following sub-sections we describe the setup and results for our classification-based GAN benchmarks. Additional details can be found in Section 5 in the Appendix.

### 3.1 DATASETS

GANs have shown promise in generating realistic samples, resulting in efforts to apply them to a broad spectrum of datasets. However, the Large-scale CelebFaces Attributes (CelebA) (Liu et al., 2015) and Large-Scale Scene Understanding (LSUN) (Yu et al., 2015) datasets remain the most popular and canonical ones in developing and evaluating GAN variants. Conveniently, these datasets also have rich annotations, making them particularly suited for our classification–based evaluations. Details on the setup for classification tasks for these datasets are given in the Appendix (Section 5).

### 3.2 MODELS

Using our framework, we perform a comparative study of several popular variants of GANs:
1. Deep Convolutional GAN (DCGAN): Convolutional GAN trained using a Jensen–Shannon divergence–based objective (Goodfellow et al., 2014; Radford et al., 2015).
2. Wasserstein GAN (WGAN): GAN that uses a Wasserstein distance–based objective (Arjovsky et al., 2017).
3. Adversarially Learned Inference (ALI): GAN that uses joint adversarial training of generative and inference networks (Dumoulin et al., 2017).
4. Boundary Equilibrium GAN (BEGAN): Auto-encoder style GAN trained using Wasserstein distance objective (Berthelot et al., 2017).

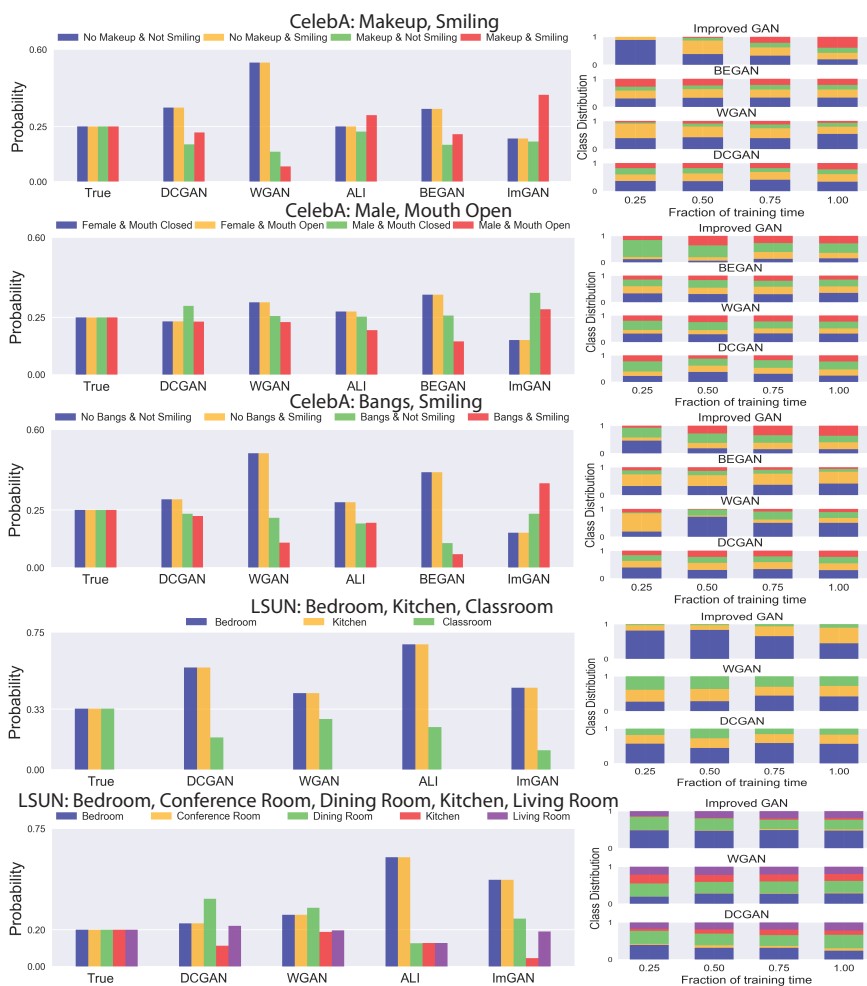

Figure 1: Visualizations of mode collapse in the synthetic, GAN-generated data produced after training on our chosen subsets of CelebA and LSUN datasets. Left panel shows the relative distribution of classes in samples drawn from synthetic datasets extracted at the end of the training process, and compares is to the true data distribution (leftmost plots). On the right, shown is the evolution of analogous class distribution for different GANs over the course of training. BEGAN did not converge on the LSUN tasks and hence is excluded from the corresponding analysis.

5. Improved GAN (ImGAN): GAN that uses semi-supervised learning (labels are part of GAN training), with various other architectural and procedural improvements (Salimans et al., 2016).

All the aforementioned GANs are unconditional, however, ImGAN has access to class labels as a part of the semi-supervised training process. We use standard implementations for each of these models, details of which are provided in the Appendix (Section 5). We also used the prescribed hyper-parameter settings for each GAN, including number of iterations we train them for. Our analysis is based on $64 \times 64$ samples, which is a size at which GAN generated samples tend to be of high quality. We also use visual inspection to ascertain that the perceptual quality of GAN samples in our experiments is comparable to those reported in previous studies. We demonstrate sample images in Figures 2 and 3 in the Appendix. BEGAN did not converge in our experiments on the LSUN dataset and hence is excluded from the corresponding analysis.

In our study, we use two types of classification models:

1. ResNet: 32-Layer Residual network He et al. (2016). Here, we choose a ResNet as it is a standard classifier in vision and yields high accuracy on various datasets, making it a reliable baseline.

2. Linear Model: This is a network with one-fully connected layer between the input and output (no hidden layers) with a softmax non-linearity. If the dimensions of input $x$ and output $\hat{y}$, are $D$ and $C$ (number of classes) respectively, then linear models implement the function $\hat{y} = \sigma(W^T x + b)$,

where $W$ is a $D \times C$ matrix, $b$ is a $C \times 1$ vector and $\sigma(\cdot)$ is the softmax function. Due to it's simplicity, this model will serve as a useful baseline in some of our experiments.

We always train the classifiers to convergence, with decaying learning rate and no data augmentation.

### 3.3 EXAMINATION OF MODE COLLAPSE

Experimental results for quantifying mode collapse through classification tasks, described in Section 2.1, are presented below. Table 2 in the Appendix gives details on datasets (subsets of CelebA and LSUN) used in our analysis, such as size ($N$), number of classes ($C$), and accuracy of the *annotator*, i.e., a classifier pre-trained on true data, which is then used to label the synthetic, GAN-generated data. Figure 1 presents class distribution in synthetic data, as specified by these annotators. The left panel compares the relative distribution of modes in true data (uniform) with that in various GAN-generated datasets. Each of these datasets is created by drawing $N$ samples from the GAN after it was trained on the corresponding true dataset. The right panel illustrates the evolution of class distributions in various GANs over the course of training[2].

**Results:** These visualization lead to the following findings:

- All GANs seem to suffer from significant mode-collapse. This becomes more apparent when the annotator granularity is increased, by considering a larger set of classes. For instance, one should compare the relatively balanced class distributions in the 3-class LSUN task to the near-absence of some modes in the 5-class task.
- Mode collapse is prevalent in GANs throughout the training process, and does not seem to recede over time. Instead the dominant mode(s) often fluctuate wildly over the course of the training.
- For each task, often there is a common set of modes onto which the distint GANs exhibit collapse.

In addition to viewing our method as an approach to analyze the mode collapse, we can also use it as a benchmark for GAN comparison. From this perspective, we can observe that, on CelebA, DCGAN and ALI learn somewhat balanced distributions, while WGAN, BEGAN and Improved GAN show prominent mode collapse. This is in contrast to the results obtained LSUN, where, for example, WGAN exhibit relatively small mode collapse, while ALI suffers from significant mode collapse even on the simple 3-class task. This highlights the general challenge in real world applications of GANs: they often perform well on the datasets they were designed for (e.g. ALI on CelebA and WGAN on LSUN), but extension to new datasets is not straightforward. Temporal analysis of mode-collapse shows that there is wide variation in the dominant mode for WGAN and Improved GAN, whereas for BEGAN, the same mode(s) often dominates the entire training process.

### 3.4 DIVERSITY EXPERIMENTS

Using the procedure outlined in Section 2.2, we perform a quantitative assessment of sample diversity in GANs on the CelebA and LSUN datasets. We restrict our experiments to binary classification as we find they have sufficient complexity to highlight the disparity between true and synthetic data. Selected results for classification-based evaluation of GANs are presented in Table 1. A more extensive study is presented in Table 3, and Figures 4 and 5 in the Appendix (Section 5).

As a preliminary check, we inspect the quality of our labeled GAN datasets. For this, we use high-accuracy *annotators* from Section 2.1 to predict labels for GAN generated data and measure consistency between the predicted and default labels (label correctness). We also inspect confidence scores, defined as the softmax probabilities for predicted class, of the annotator. The motivation behind these metrics is that if the classifier can correctly and with high-confidence predict labels for labeled GAN samples, then it is likely that they are convincing examples of that class, and hence of good "quality". Empirical results for label agreement and annotator confidence of GAN generated datasets are shown in Tables 1 and 3, and Figure 4. We also report an equivalent Inception Score (Salimans et al., 2016), similar to that described in Section 2.3. Using the Inception network to get the label distribution may not be meaningful for face or scene images. Instead, we compute the Inception Score using the label distribution predicted from the annotator networks. Score is computed as $exp(\mathbb{E}_x[\mathbf{KL}(p(y|x))||p(y)])$, where $y$ refers to label predictions from the annotators [3].

---

[2]Temporal evaluation of ALI class distribution is absent in the analysis due to absence of periodic check-pointing provisions in the code.

[3]Code from `https://github.com/openai/improved-gan`

| Task | Data Source | Classification Performance | | | | | | |
|---|---|---|---|---|---|---|---|---|
| | | Label Correctness (%) | Inception Score ($\mu \pm \sigma$) | Accuracy (%) | | | | |
| | | | | Linear model | | | | ResNet |
| | | | | $\uparrow_1$ | | $\uparrow_{10}$ | | $\uparrow_1$ |
| | | | | Train | Test | Train | Test | Test |
| CelebA Smiling (Y/N) # Images: 156160 | True | | | 85.7 | 85.6 | | | 92.4 |
| | True $\downarrow_{64}$ | | | 87.6 | 85.0 | | | 87.8 |
| | True $\downarrow_{256}$ | 92.4 | $1.69 \pm 0.0074$ | 91.5 | 82.4 | | | 82.1 |
| | True $\downarrow_{512}$ | | | 93.7 | 80.0 | | | 77.8 |
| | True $\downarrow_{1024}$ | | | 95.0 | 76.2 | | | 71.2 |
| | DCGAN | 96.1 | $1.67 \pm 0.0028$ | 100.0 | 77.1 | 100.0 | 77.1 | 63.3 |
| | WGAN | 98.2 | $1.68 \pm 0.0031$ | 96.8 | 83.4 | 96.8 | 83.5 | 65.3 |
| | ALI | 93.3 | $1.71 \pm 0.0027$ | 94.5 | 80.1 | 95.0 | 82.4 | 55.8 |
| | BEGAN | 93.5 | $1.74 \pm 0.0028$ | 98.5 | 69.5 | 98.5 | 69.6 | 64.1 |
| | Improved GAN | 98.4 | $1.88 \pm 0.0021$ | 100.0 | 70.2 | 100.0 | 70.1 | 61.6 |
| LSUN Bedroom/Kitchen # Images: 200000 | True | | | 64.7 | 64.1 | | | 99.1 |
| | True$\downarrow_{512}$ | | | 64.7 | 64.0 | | | 76.4 |
| | True $\downarrow_{1024}$ | 98.2 | $1.94 \pm 0.0217$ | 65.2 | 64.0 | | | 66.9 |
| | True $\downarrow_{2048}$ | | | 98.7 | 56.2 | | | 56.5 |
| | True $\downarrow_{4096}$ | | | 100.0 | 55.1 | | | 55.1 |
| | DCGAN | 92.7 | $1.85 \pm 0.0036$ | 90.8 | 56.5 | 91.2 | 56.3 | 51.2 |
| | WGAN | 87.8 | $1.70 \pm 0.0023$ | 86.2 | 58.2 | 96.3 | 54.1 | 55.7 |
| | ALI | 80.4 | $1.62 \pm 0.0026$ | 80.7 | 49.7 | 81.7 | 50.8 | 50.5 |
| | Improved GAN | 84.2 | $1.68 \pm 0.0030$ | 91.6 | 55.9 | 90.8 | 56.5 | 51.2 |

Table 1: Select results from the comparative study on classification performance of true data vs. GANs on the CelebA and LSUN datasets. Label correctness measures the agreement between default labels for the synthetic datasets, and those predicted by the annotator, a classifier trained on true data. Shown alongside are the equivalent inception scores computed using labels predicted by the annotator (rather than an Inception Network). Training and test accuracies for a linear model on the various true and synthetic datasets are reported. Also presented are the corresponding accuracies for this classifier trained on down-sampled true data ($\downarrow_M$) and oversampled synthetic data ($\uparrow_L$). Test accuracy for ResNets trained on these datasets is also shown (training accuracy was always $100\%$), though it is noticeable that deep networks suffer from issues when trained on synthetic datasets.

Next, we train classifiers using the true and labeled GAN-generated datasets and study their performance in terms of accuracy on a hold-out set of true data. ResNets (and other deep variants) yield good classification performance on true data, but suffer from severe overfitting on the synthetic data, leading to poor test accuracy. This already indicates a possible problem with GANs and the diversity of the data they generate. But to highlight this problem better and avoid the issues that stem from overfitting, we also look for a classifier which does not always overfit on the synthetic data. We, however, observed that even training simple networks, such as one fully connected layer with few hidden units, led to overfitting on synthetic data. Hence, we resorted to a very basic *linear* model described in Section 3.2. Tables 1 and 3 shows results from binary classification experiments using linear models, with the training and test accuracies of the classifier on various datasets.

Finally, to get a better understanding of the underlying "diversity" of synthetic datasets, we train linear models using down-sampled versions of true data (no augmentation), and compare this to the performance of synthetic data, as shown in Tables 1 and 3. Down-sampling the data by a factor of $M$, denoted as $\downarrow_M$ implies selecting a random $N/M$ subset of the data $D$. Visualizations of how GAN classification performance compares with (down-sampled) true data are in Figure 5 in the Appendix. A natural argument in the defense of GANs is that we can oversample them, i.e. generate datasets much larger than the size of training data. Results for linear models trained using a 10-fold oversampling of GANs (drawing $10N$ samples), denoted by $\uparrow_{10}$, are show in Tables 1 and 3.

**Results:** The major findings from these experiments are:
- Based on Tables 1 and 3, and Figure 4, we see strong agreement between annotator labels and true labels for synthetic data, on par with the scores for the test set of true data. It is thus apparent that the GAN images are of high-quality, as expected based on the visual inspection. These scores are lower for LSUN than CelebA, potentially due to lower quality of generated LSUN

images. From these results, we can get a broad understanding of how good GANs are at producing convincing/representative samples from different classes across datasets. This also shows that simple classification-based benchmarks can highlight relevant properties of synthetic datasets.

- The equivalent inception score is not very informative and is similar for the true (hold-out set) and synthetic datasets. This is not surprising given the simple nature of our binary classification task and the fact that the true and synthetic datasets have almost a uniform distribution over labels.

- It is evident from Table 1 that there is a large performance gap between true and synthetic data on classification tasks. Inspection of training accuracies shows that linear models are able to nearly fit the synthetic datasets, but are grossly underfitting on true data. Given the high scores of synthetic data on the previous experiments to assess dataset 'quality' (Tables 1 and 3, and Figure 4), it is likely that the poor classification performance is more indicative of lack of 'diversity'.

- Comparing GAN performance to that of down-sampled true data reveals that the learned distribution, which was trained on datasets that have *around hundred thousand* data points exhibits diversity that is on par with what *only mere couple of hundreds* of true data samples constitute! This shows that, at least from the point of view of classification, the diversity of the GAN generated data is severely lacking.

- Oversampling GANs by 10-fold to produce larger datasets does not improve classification performance. The disparity between true and synthetic data remains nearly unchanged even after this significant oversampling, further highlighting the lack of diversity in GANs.

In terms of the conclusions of relative performance of various GANs, we observe that WGAN and ALI (on CelebA) perform better than the other GANs. While BEGAN samples have good perceptual quality (see Figure 2), they consistently perform badly on our classification tasks. On the other hand, WGAN samples have relatively poor visual quality but seem to outperform other GANs in classification tasks. This is a strong indicator of the need to consider other metrics, such as the ones proposed in this paper, in addition to visual inspection to study GANs. For LSUN, the gap between true and synthetic data is much larger, with the classifiers getting near random performance on all the synthetic datasets. Note that these classifiers get poor test accuracy on LSUN but are not overfitting on the training data. In this case, we speculate the lower performance could be due to both lower quality and diversity of LSUN samples.

In summary, our key experimental finding is that even simple classification–based tests can hold tremendous potential to shed insight on the learned distribution in GANs. This not only helps us to get a deeper understanding of many of the underlying issues, but also provides with a more quantitative and rigorous platform on which to compare different GANs. Our techniques could, in principle, be also applied to assess other generative models such as Variational Auto-Encoders (VAEs) Kingma & Welling (2014). However, VAEs have significant problems in generating realistic samples on the datasets used in our analysis in the first place – see Arora & Zhang (2017).

## 4    CONCLUSIONS

In this paper, we put forth techniques for examining the ability of GANs to capture key characteristics of the training data, through the lens of classification. Our tools are scalable, quantitative and automatic (no need for visual inspection of images). They thus are capable of studying state-of-the-art GANs on realistic, large-scale image datasets. Further, they serve as a mean to perform a nuanced comparison of GANs and to identify their relative merits, including properties that cannot be discerned from mere visual inspection.

We then use the developed techniques to perform empirical studies on popular GANs on the CelebA and LSUN datasets. Our examination shows that mode collapse is indeed a prevalent issue for GANs. Also, we observe that synthetic GAN-generated datasets have significantly reduced diversity, at least when examined from a classification perspective. In fact, the diversity of such synthetic data is often few orders of magnitude smaller than that of the true data. Furthermore, this gap in diversity does not seem to be bridged by simply producing much larger datasets by oversampling GANs. Finally, we also notice that good perceptual quality of samples does not necessarily correlate – and might sometime even anti-correlate – with distribution diversity. These findings suggest that we need to go beyond the visual inspection–based evaluations and look for more quantitative tools for assessing quality of GANs, such as the ones presented in this paper.

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

## 5 APPENDIX

### 5.1 EXPERIMENTAL SETUP

#### 5.1.1 DATASETS FOR CLASSIFICATION TASKS

To assess GAN performance from the perspective of classification, we construct a set of classification tasks on the CelebA and LSUN datasets. In the case of the LSUN dataset, images are annotated with scene category labels, which makes it straightforward to use this data for binary and multi-class classification. On the other hand, each image in the CelebA dataset is labeled with 40 binary attributes. As a result, a single image has multiple associated attribute labels. Here, we construct classification tasks can by considering binary combinations of an attribute(s) (examples are shown in Figure 2). Attributes used in our experiments were chosen such that the resulting dataset was large, and classifiers trained on true data got high-accuracy so as to be good *annotators* for the synthetic data. Details on datasets used in our classification tasks, such as training set size ($N$), number of classes ($C$), and accuracy of the annotator, i.e., a classifier pre-trained on true data which is used to label the synthetic GAN-generated data, are provided in Table 2.

| Dataset | $N$ | $C$ | Annotator's Accuracy (%) |
|---|---|---|---|
| CelebA: Makeup, Smiling | 102,436 | 4 | 90.9, 92.4 |
| CelebA: Male, Mouth Open | 115,660 | 4 | 97.9, 93.5 |
| CelebA: Bangs, Smiling | 45,196 | 4 | 93.9, 92.4 |
| LSUN: Bedroom, Kitchen, Classroom | 150,000 | 3 | 98.7 |
| LSUN: Bedroom, Conference Room, Dining Room, Kitchen, Living Room | 250,000 | 5 | 93.7 |

Table 2: Details of CelebA and LSUN subsets used for the studies in Section 3.3. Here, we use a classifier trained on true data as an *annotator* that let's us infer label distribution for the synthetic, GAN-generated data. $N$ is the size of the training set and $C$ is the number of classes in the true and synthetic datasets. Annotator's accuracy refers to the accuracy of the classifier on a test set of true data. For CelebA, we use a combination of attribute-wise binary classifiers as annotators due their higher accuracy compared to a single classifier trained jointly on all the four classes.

#### 5.1.2 MODELS

Benchmarks were performed on standard implementations -

- DCGAN: `https://github.com/carpedm20/DCGAN-tensorflow`
- WGAN: `https://github.com/martinarjovsky/WassersteinGAN`
- ALI: `https://github.com/IshmaelBelghazi/ALI`
- BEGAN :`https://github.com/carpedm20/BEGAN-tensorflow`
- Improved GAN: `https://github.com/openai/improved-gan`
- ResNet Classifier: Variation of the standard TensorFlow ResNet `https://github.com/tensorflow/models/blob/master/research/resnet/resnet_model.py`

### 5.2 ADDITION EXPERIMENTAL RESULTS

#### 5.2.1 SAMPLE QUALITY

For each of our benchmark experiments, we ascertain that the visual quality of samples produced by the GANs is comparable to that reported in prior work. Examples of random samples drawn for multi-class datasets from both true and synthetic data are shown in Figure 2 for the CelebA dataset, and in Figure!3 for the LSUN dataset.

#### 5.2.2 MODE COLLAPSE EXPERIMENTS

In the studies to observe mode collapse in GANs described in Sections 2.1 and 3.3, we use a pre-trained classifier as an annotator to obtain the class distribution for datasets generated from unconditional GANs. Figure 4 shows histograms of annotator confidence for the datasets used for benchmarking listed in Table 2. As can be seen in these figures, the annotator confidence for the synthetic data is comparable to that on the hold-out set of true data. Thus, it seems likely that the

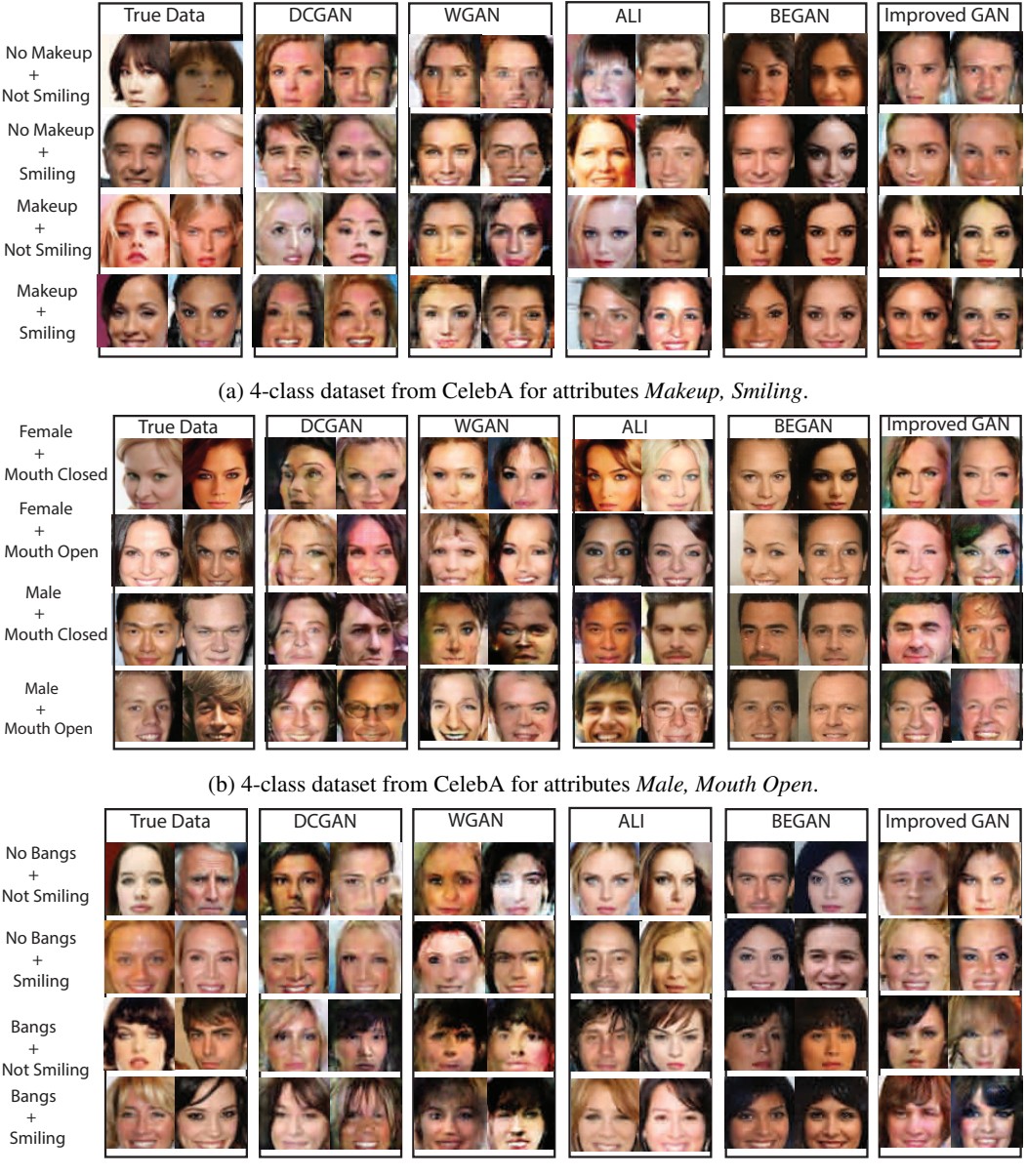

(a) 4-class dataset from CelebA for attributes *Makeup, Smiling*.

(b) 4-class dataset from CelebA for attributes *Male, Mouth Open*.

(c) 4-class dataset from CelebA for attributes *Bangs, Smiling*.

Figure 2: Illustration of datasets from CelebA used in proposed classification-based benchmarks to evaluate GANs. Shown alongside are images sampled from various unconditional GANs trained on this dataset. Labels for the GAN samples are obtained using a pre-trained classifier as an annotator.

GAN generated samples are of good quality and are truly representative examples of their respective classes, as expected based on visual inspection.

### 5.2.3 DIVERSITY EXPERIMENTS

Table 3 presents an extension of the comparative study of classification performance of true and GAN generated data provided in Table 1. Visualizations of how test accuracies of a linear model classifier trained on GAN data compares with one trained on true data is shown in Figure 5. For each task, the bold curve shows test accuracy of a classifier trained on true data as a function of true dataset size. A down-sampling factor of $M$ corresponds to training the classifier on a random $N/M$ subset of true data. The dashed curves show test accuracy of classifiers trained on GAN datasets,

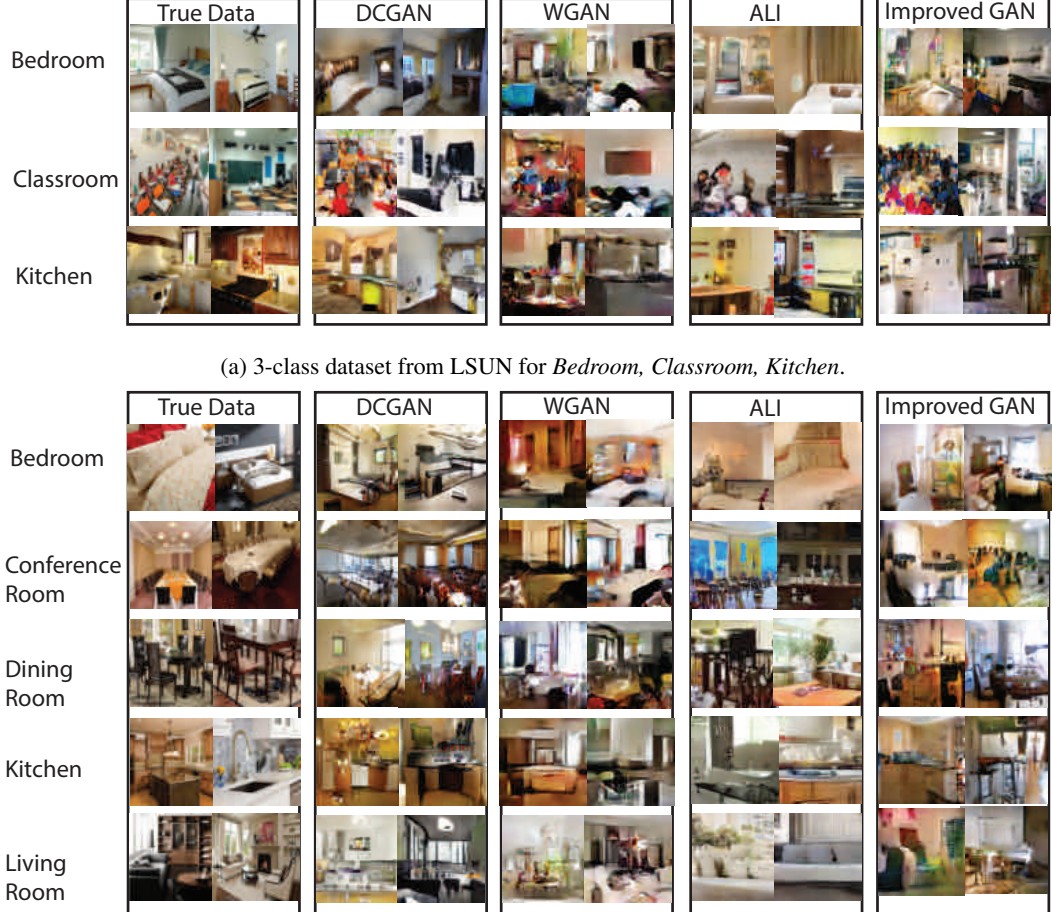

(a) 3-class dataset from LSUN for *Bedroom, Classroom, Kitchen*.

(b) 5-class dataset from LSUN for *Bedroom, Conference Room, Dining Room, Kitchen, Living Room*.

Figure 3: Illustration of datasets from LSUN used in proposed classification-based benchmarks to evaluate GANs. Shown alongside are images sampled from various unconditional GANs trained on this dataset. Labels for the GAN samples are obtained using a pre-trained classifier as an annotator.

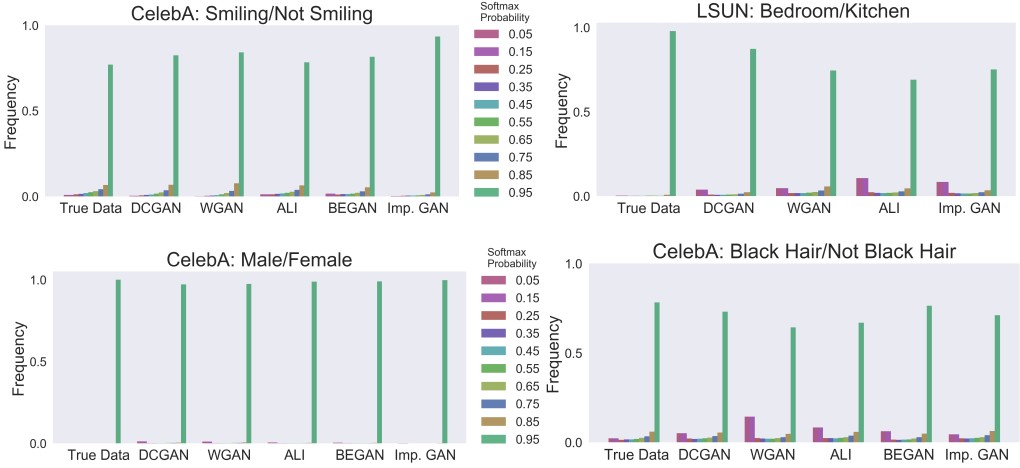

Figure 4: Histograms of annotator confidence (softmax probability) during label prediction on true data (test set) and synthetic data for tasks on the CelebA and LSUN datasets (see Section 3.4).

obtained by drawing $N$ samples from GANs at the culmination of the training process. Based on these visualizations, it is apparent that GANs have comparable classification performance to a subset of training data that is more than a 100x smaller. Thus, from the perspective of classification, GANs have diversity on par with a few hundred true data samples.

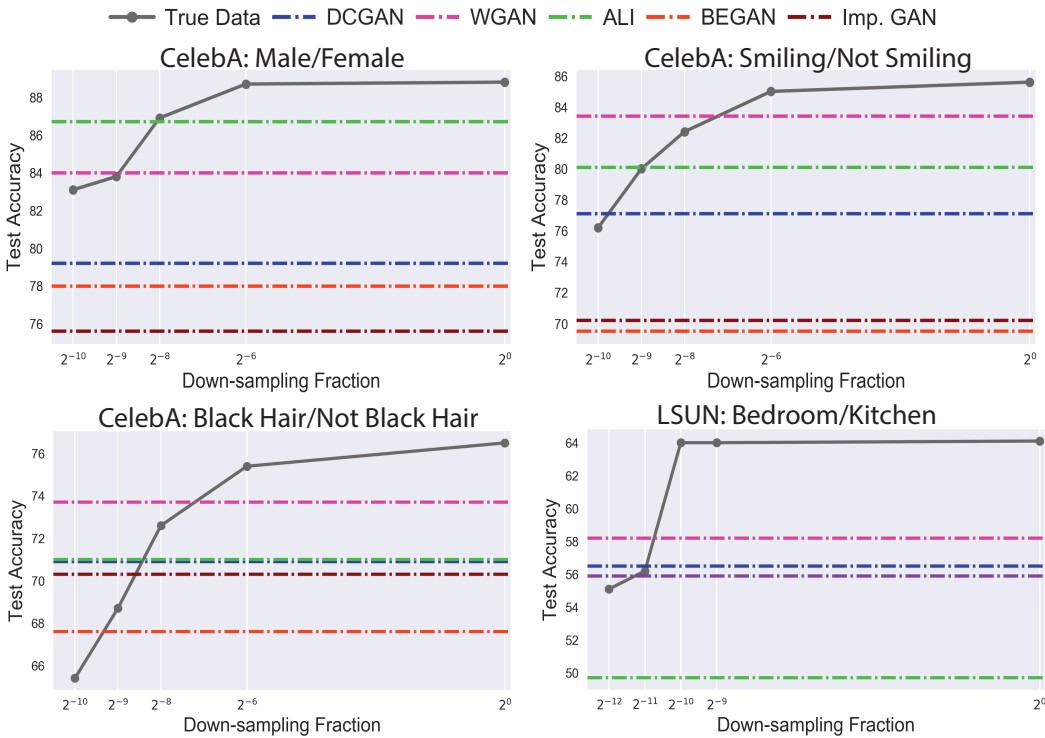

Figure 5: Illustration of the classification performance of true data compared with GAN-generated synthetic datasets based on experiments described in Section 3.4. Classification is performed using a basic linear model, described in Section 3.2, and performance is reported in terms of accuracy on a hold-out set of true data. In the plots, the bold curve captures the classification performance of models trained on true data vs the size of the true dataset (maximum size is $N$). Dashed lines represent performance of classifiers trained on various GAN-generated datasets of size $N$. These plots indicate that GAN samples have "diversity" comparable to a small subset (few hundred samples) of true data. Here the notion of diversity is one that is relevant for classification tasks.

| Task | Data Source | Label Correctness (%) | Inception Score ($\mu \pm \sigma$) | Accuracy (%) | | | | |
|---|---|---|---|---|---|---|---|---|
| | | | | Linear model | | | | ResNet |
| | | | | $\uparrow_1$ | | $\uparrow_{10}$ | | $\uparrow_1$ |
| | | | | Train | Test | Train | Test | Test |
| CelebA Male (Y/N) # Images: 136522 | True | 97.9 | $1.98 \pm 0.0033$ | 88.1 | 88.8 | | | 97.9 |
| | True $\downarrow_{64}$ | | | 89.6 | 88.7 | | | 92.9 |
| | True $\downarrow_{256}$ | | | 91.6 | 86.9 | | | 89.8 |
| | True $\downarrow_{512}$ | | | 96.3 | 83.8 | | | 82.6 |
| | True $\downarrow_{1024}$ | | | 100.0 | 83.1 | | | 81.4 |
| | DCGAN | 98.2 | $1.97 \pm 0.0013$ | 100.0 | 79.2 | 100.0 | 79.6 | 56.4 |
| | WGAN | 98.3 | $1.97 \pm 0.0013$ | 96.7 | 84.0 | 96.7 | 83.9 | 50.0 |
| | ALI | 99.2 | $1.99 \pm 0.0008$ | 95.8 | 86.7 | 95.8 | 86.7 | 58.9 |
| | BEGAN | 99.3 | $1.99 \pm 0.0006$ | 97.9 | 78.0 | 98.0 | 78.2 | 55.4 |
| | Improved GAN | 99.8 | $1.99 \pm 0.0004$ | 100.0 | 75.6 | 100.0 | 71.0 | 71.7 |
| CelebA Smiling (Y/N) # Images: 156160 | True | 92.4 | $1.69 \pm 0.0074$ | 85.7 | 85.6 | | | 92.4 |
| | True $\downarrow_{64}$ | | | 87.6 | 85.0 | | | 87.8 |
| | True $\downarrow_{256}$ | | | 91.5 | 82.4 | | | 82.1 |
| | True $\downarrow_{512}$ | | | 93.7 | 80.0 | | | 77.8 |
| | True $\downarrow_{1024}$ | | | 95.0 | 76.2 | | | 71.2 |
| | DCGAN | 96.1 | $1.67 \pm 0.0028$ | 100.0 | 77.1 | 100.0 | 77.1 | 63.3 |
| | WGAN | 98.2 | $1.68 \pm 0.0031$ | 96.8 | 83.4 | 96.8 | 83.5 | 65.3 |
| | ALI | 93.3 | $1.71 \pm 0.0027$ | 94.5 | 80.1 | 95.0 | 82.4 | 55.8 |
| | BEGAN | 93.5 | $1.74 \pm 0.0028$ | 98.5 | 69.5 | 98.5 | 69.6 | 64.1 |
| | Improved GAN | 98.4 | $1.88 \pm 0.0021$ | 100.0 | 70.2 | 100.0 | 70.1 | 61.6 |
| CelebA Black Hair (Y/N) # Images: 77812 | True | 84.5 | $1.68 \pm 0.0112$ | 76.4 | 76.5 | | | 84.5 |
| | True $\downarrow_{64}$ | | | 79.7 | 75.4 | | | 80.0 |
| | True $\downarrow_{256}$ | | | 86.3 | 72.6 | | | 75.8 |
| | True $\downarrow_{512}$ | | | 89 | 68.7 | | | 73.9 |
| | True $\downarrow_{1024}$ | | | 100.0 | 65.4 | | | 72.7 |
| | DCGAN | 86.7 | $1.68 \pm 0.0040$ | 100.0 | 70.9 | 100.0 | 70.5 | 53.4 |
| | WGAN | 76.0 | $1.60 \pm 0.0055$ | 94.4 | 73.7 | 94.3 | 73.4 | 58.5 |
| | ALI | 79.4 | $1.63 \pm 0.0028$ | 94.9 | 71.0 | 94.9 | 70.2 | 55.7 |
| | BEGAN | 87.6 | $1.74 \pm 0.0028$ | 94.1 | 67.6 | 94.1 | 67.7 | 67.2 |
| | Improved GAN | 86.7 | $1.64 \pm 0.0045$ | 100.0 | 70.3 | 100.0 | 69.1 | 70.2 |
| LSUN Bedroom/Kitchen # Images: 200000 | True | 98.2 | $1.94 \pm 0.0217$ | 64.7 | 64.1 | | | 99.1 |
| | True$\downarrow_{512}$ | | | 64.7 | 64.0 | | | 76.4 |
| | True $\downarrow_{1024}$ | | | 65.2 | 64.0 | | | 66.9 |
| | True $\downarrow_{2048}$ | | | 98.7 | 56.2 | | | 56.5 |
| | True $\downarrow_{4096}$ | | | 100.0 | 55.1 | | | 55.1 |
| | DCGAN | 92.7 | $1.85 \pm 0.0036$ | 90.8 | 56.5 | 91.2 | 56.3 | 51.2 |
| | WGAN | 87.8 | $1.70 \pm 0.0023$ | 86.2 | 58.2 | 96.3 | 54.1 | 55.7 |
| | ALI | 80.4 | $1.62 \pm 0.0026$ | 80.7 | 49.7 | 81.7 | 50.8 | 50.5 |
| | Improved GAN | 84.2 | $1.68 \pm 0.0030$ | 91.6 | 55.9 | 90.8 | 56.5 | 51.2 |

Table 3: Detailed version of the comparative study of the classification performance of true data and GANs on the CelebA and LSUN datasets shown in Table 1, based on experiments described in Section 3.4. Label correctness measures the agreement between default labels for the synthetic datasets, and those predicted by the annotator, a classifier trained on the true data. Shown alongside are the equivalent inception scores computed using labels predicted by the annotator (instead of the Inception Network). Training and test accuracies for a linear model classifier on the various true and synthetic datasets are reported. Also presented are the corresponding accuracies for a linear model trained on down-sampled true data ($\downarrow_M$) and oversampled synthetic data ($\uparrow_L$). Test accuracy for ResNets trained on these datasets is also shown (training accuracy was always $100\%$), though it is noticeable that deep networks suffer from issues when trained on synthetic datasets.

