# OpenReview forum: "A Classification-Based Perspective on GAN Distributions"
_ICLR.cc/2018/Conference — Reject_

### Official Review · AnonReviewer3 · 2017-11-19
**Trying to shed light at comparison between different GAN variants, but the metrics introduced are not very novel, results are not comparable with prior work and older version of certain models are used**

**Rating:** 5
**Confidence:** 5

**Review:**

Overall comments: Trying to shed light at comparison between different GAN variants, but the metrics introduced are not very novel, results are not comparable with prior work and older version of certain models are used (WGAN instead of Improved WGAN)

Section 2.1: quantifying mode collapse
* This section should mention Inception scores. A model which collapses on only one class will have a low inception score, and this metric also uses a conv net classifier, as the approach is very similar (the method is only mentioned briefly in section 2.3)
* The authors might not be aware of concurrent work published before the ICLR deadline, which introduces a very similar metric: https://arxiv.org/abs/1706.08500

Section 2.2: measuring diversity:
* There is an inherent flaw in this metric, namely it trains one GAN per class. One cannot generalize from this metric on how different GAN models will perform when trained on the entire dataset. One model might be able to capture more diverse distributions, but lose a bit of quality, while another model might be able to create good samples when train on low diversity data. We already know that when looking at other generative models, we can find such examples. VAEs can obtain very good samples on celebA, a dataset with relative low diversity, but not so good samples on cifar.
* The authors compare their experiment with Radford et al. (2015), but that needs to be done with caution. In Radford et al. (2015), the authors use a conditional generative model trained on the entire dataset. In that setting, this test is more suitable since one can test how good well the model has learned the conditioning. For example, for a conditional model trained on cats and dogs, a failure mode is that the model generates only cats. This failure mode can then be captured by this metric. However, when training two models, one on cats and one on dogs, this failure mode is not present since the data is already split into classes.
* The proposed metric is not necessarily a diversity metric, it is also a quality metric:  in a situation where all the models diverge and generate random noise, with high diversity, but without any structure. This metric would capture this issue, because a classifier will not be able to learn the classes, because there is no correlation between the classes and the generated images.

Experimental results:
* Positive insights regarding labels and celeba. Looks like subtle labels on faces are not being captured by GAN models.
Figure 1 is hard to read.
* ALI having higher diversity on celeba is explicitly mentioned in a paper the authors cite, namely “Do GANs actually learn the distribution? An empirical study”. Would be nice to mention that in the paper.

Would like to see:
* A comparison with the Improved Wasserstein GAN model. This model is now the one used by the community, as opposed to the original Wasserstein GAN.
* Models trained on cifar, with the reported inception scores of the models on cifar. That makes the paper comparable with previous work and is a test against bugs in model implementations or other parts of the code. This would also allow to test for claims such as the fact that the Improved GAN has more mode collapse than DCGAN, while the Improved GAN paper says the opposite.
The reason why the authors chose the models they did for comparison. In the BiGAN (same model as ALI) paper the authors report a low inception score, which suggests that their model is not able to capture the subtleties of the Cifar dataset, and this seems to be correlated with the results obtained in this work.

---

> ### Author Response · Authors · 2018-01-05
> **Authors' Response**
>
> We thank the reviewer for their analysis and questions. Below we address specific concerns raised by the reviewer.
>
> Section 2.1: quantifying mode collapse
> We compare to the Inception Score in all experiments performed in Section 3.4 (Tables 1 and 3) and will include a similar comparison for the mode collapse experiments in the paper revision. However, we believe that the Inception Score has certain shortcomings for measuring diversity, as discussed in point 3 of our response to AnonReviewer2. We thank the reviewer pointing us to the concurrent work of Heusel et al. (2017). We will compare to it in the paper revision.
>
> Section 2.2: measuring diversity:
> * We would like to clarify the premise of our metric - if a GAN trained per class learned the true distribution for that class, then samples from these class-wise GANs would be able to accurately reconstruct the inter-class decision boundaries. Given the reduced performance of GANs on classification tasks when compared to true data, even for binary classification, it is clear that the GAN distributions are inherently lacking.
>
> * We agree with the reviewer that if a GAN is trained separately for each class, one cannot identify mode dropping (at the scale of whole classes) based on the resulting samples. However, we would like to clarify that this is the approach we propose for diversity studies, where we assess how good GANs are at classification (and do not attempt to directly measure mode collapse). In such a setting, we want to have balanced datasets, with an equal number of examples per class.
>
> Our approach to study mode dropping is explained in Sections 2.1 and 3.3. This method does not involve training a separate GAN per class. In fact, a single GAN is trained on a multi-class dataset and then a pre-trained classifier is used as an annotator to observe learned modes. The approach proposed in Radford et al. (2015) uses a conditional GAN, wherein sampling requires specifying the class label, e.g., cat or dog. Thus it is unlikely that these GANs would generate only cats since the images are produced conditioned on the label - similar to sampling from class-wise GANs. Hence we believe that the GANs used in the Radford et al. (2015) study would be less likely to exhibit dropping of entire classes as compared to a GAN trained unconditionally. Thus the later is deliberately chosen for our mode collapse studies.
>
> * We agree with the reviewer that classification performance is a measure of both quality and diversity. However, in the setting of common GANs, images are known to have good perceptual quality. This prior is reinforced by the high ‘confidence scores’ and Inception scores observed in our experiments for the synthetic GAN samples (Section 3.4). Thus we believe that the reduced classification performance is more a result of lower diversity than quality.
>
> Experimental results:
>
> * We will make Figure 1 more readable in the paper revision. We will also include the reference to Arora & Zhang (2017) in the comment on ALI diversity.
>
> * In this paper, we tried to identify GANs that are somewhat classic (e.g., DCGAN and WGAN) and some newer GANs to evaluate. We also chose GANs that have been studied by similar prior work (Arora & Zhang (2017)). Our objective was not to provide a relative ranking of all existing GANs but to propose a methodology to perform large-scale automated analysis of the learned image distributions in GANs. So, our choice of GANs was focused on demonstrating this methodology on popular variants. It is straightforward to apply this suite of tools to any other GAN, and we will include an evaluation of Improved WGAN in the paper revision.
>
> “Would like to see”:
>
> In this paper, we use standard implementations for all GANs as mentioned in Section 5.1.2. We tried to use CIFAR-10 as a dataset in our evaluations. However, in our experiments, most of the GANs did not generate meaningful images on this dataset, as these GANs have been not optimized for this setting. Thus, we decided to use datasets for which these GANs have been optimized - CelebA and LSUN.
>
> However, as an additional sanity check of our setup, below we provide results for DCGAN trained on CIFAR-10 (as this GAN produces realistic samples on CIFAR-10). Our experiment was as follows:
> 1. We trained an unconditional DCGAN on CIFAR-10. The Inception Score measured on 50000 GAN samples is 6.5104±0.0698087, similar to prior art (https://github.com/xunhuang1995/SGAN).
> 2. We trained 10 class-wise DCGANs on CIFAR-10. The Inception Score measured on 50000 GAN samples, 5000 from each class-wise GAN is 5.94101±0.051225. We performed diversity studies (Section 3.4) on these samples. A ResNet trained on the true CIFAR-10 without data augmentation gets test accuracy of 85.2% after 60k steps whereas on DCGAN samples it gets near-random performance of 17.4% (training accuracy is 100%).
>
> We will include a comparison of DCGAN and Improved GAN on CIFAR-10 in the paper revision.

---

### Official Review · AnonReviewer1 · 2017-11-28
**Training a classifier to evaluate GANs**

**Rating:** 6
**Confidence:** 4

**Review:**

The paper proposes a new evaluation measure for evaluating GANs. Specifically, the paper proposes generating synthetic images using GAN, training a classifier (for an auxiliary task, not the real vs fake discriminator) and measuring the performance of this classifier on held out real data.

While the idea of using a downstream classification task to evaluate the quality of generative models has been explored before  (e.g. semi-supervised learning), I think that this is the first paper to evaluate GANs using such an evaluation metric.

I'm not super convinced that this is an useful evaluation metric as the absolute number is somewhat to interpret and dependent on the details of the classifier used. The results in Table 1 change quite a bit depending on the classifier.

It would be useful to add a discussion of the failure modes of the proposed metric. It seems like a generator which generates samples close to the classification boundary (but drops examples far away from the boundary) could still achieve a high score under this metric.

In the experiments, were different architectures used for different GAN variants?

I think the mode-collapse evaluation metrics in MR-GAN are worth discussing in Section 2.1
Mode Regularized Generative Adversarial Networks
https://arxiv.org/abs/1612.02136

---

> ### Author Response · Authors · 2018-01-05
> **Authors' Response**
>
> We thank the reviewer for their analysis and comments. Below we address specific questions and concerns:
>
> 1. We agree with the reviewer that the absolute value of the scores might depend on the choice of classifier used. This is, however, true even for the inception score, another popular metric used to evaluate GANs. (After all, the inception score directly relies on the Inception architecture).  Still, we believe that the relative performance of different GANs and, especially, how they compare to true data performance already provides sufficiently meaningful information. It is worth noting that true data gets more than 90% classification accuracy on both the simple linear classifier and a deep ResNet, whereas synthetic GAN data performance is far from comparable to true data for either of the classifiers. In our experiments, we also studied other classifier networks and observed that the trends of the relative performance of the GANs was preserved irrespective of the choice of classifier.
>
> 2. We thank the reviewer for this comment and will include a discussion of the failure modes of our approach in the revised version of the paper. It is true that a GAN could attain good classification score by generating samples close to the decision boundary. However, using the proposed mode collapse experiments we would be able to identify this phenomenon. In these experiments, we use a pre-trained classifier to annotate the data and measure dropping of modes. Here, using a classifier with really fine-grained annotation (more than just two classes), we could ascertain whether the GAN is omitting modes away from the decision boundary.
>
> More importantly, we believe that quantifying the behavior of a generative model inherently requires using multiple metrics. Given that we aim to evaluate very complex behaviour in terms of simple, concise metrics, each of these metrics individually will inevitably conflate some aspects. In particular, our diversity score, by definition, can only capture certain aspects of diversity and thus should be used in conjunction with other evaluation metrics that would be more sensitive to, e.g., a dataset concentrated around the decision boundary.
>
> 3. For every GAN, we used the standard architecture, code and hyperparameters provided by the respective authors as listed in Section 5.1.2. We did this to compare the most optimized and best-performing version of each GAN, as well as to emphasize that our methodology can be used “out of the box”, without a need for customization. For the classifiers, the same architecture, code, and hyperparameters were used for both true data and synthetic data. Thus in any column of Table 1 and Table 3, the results are based on exactly the same classifier network.
>
> 4. We thank the reviewer for this suggestion and will include the metric proposed in MR-GAN (Che et al. (2017)) as a comparison in the revised version of our paper. It is worth noting, however, that even the authors of this paper suggest that their metric may not be appropriate for the CelebA and LSUN datasets (as mentioned in our reply to AnonReviewer2, we focused on these two datasets because they are the most commonly used in the context of state-of-the-art GANs).

---

### Official Review · AnonReviewer2 · 2017-11-28
**Neat idea. But not worth publishing**

**Rating:** 3
**Confidence:** 4

**Review:**

This paper propose to evaluate the distributions learned by GAN using classification-based methods. As two examples, the authors evaluates the mode collapse effect and measure the diversity for GAN distributions. The proposed approaches are experimental but does not require human inspection. The main idea is to fit a classifier on the training data and also learn a GAN model using the training data. Then generate simulated data using GAN and use the classifier to predict the labels of the simulated data. The distribution of predicted labels  and the labels of the true data can be easily compared.

Detailed comments:

1. The proposed method is purely experimental. It  would be better to gain some theoretical insights of this methodology. Moreover, in terms of experiments, it would be nice to consider more examples except for mode collapse and diversity, since these problems are well-known for GAN.

2. Since mode collapse is a well-known phenomenon, the novelty of this paper is not sufficient.

3. There are other measures for the quality of GAN. For example, the inception scores and mode scores (Salimans et al. 2016, Che et al. 2017). It would be nice to compare the method here with other related work.

References:
1. Improved Techniques for Training GANs https://arxiv.org/abs/1606.03498

2. Mode Regularized Generative Adversarial Networks https://arxiv.org/abs/1612.02136

---

> ### Author Response · Authors · 2018-01-05
> **Authors' Response**
>
> We thank the reviewer for their comments. The reviewer accurately summarized the part of our experiments relating to mode collapse. However, we would like to remark that there is another part of our submission that provides a more general methodology for evaluating the quality of the learned distribution in GANs. It focuses on measuring fidelity of class-wise decision boundaries. We view this as an important aspect of our work and thus want to bring it to the reviewer’s attention.
>
> Before addressing the specific points raised in the review, we would like to make a broader comment:
>
> The overarching goal of generative models is to produce samples of comparable quality and diversity to the distribution they are trained on (true distribution). While many of the state-of-the-art GANs produce synthetic samples that have good perceptual quality, it is rather clear at this point that the primary challenge for the current GANs is their inability to fully capture the diversity of the true distribution (and mode collapse is just one aspect of this problem.)  This phenomenon is well known at the intuitive level but remains fairly vague (in particular, the concept of mode collapse has been so far defined concretely only in the context of very simple models such as Gaussian mixtures). It is thus crucial to make the notion of “a GAN (not) capturing the diversity of the underlying distribution” be precise and measurable. Otherwise, it will be difficult to make progress in this area in a principled manner.
>
> The primary goal of our work is to address this exact challenge. Specifically, we view our key contribution to be putting forth a methodology to enable us to measure the “learned diversity” of GAN distributions in a quantitative manner and, crucially, to make this methodology be fully scalable and automated (in particular, to make it not require visual inspection by a human). This way, the resulting benchmarks are directly applicable to actual state-of-the-art GANs on realistic high-dimensional image datasets (instead of having to rely solely on toy models and/or datasets). They can thus be used to, on one hand, obtain a much more fine-grained understanding of the distributions that actual state-of-the-art GANs learn and, on the other hand, to guide further development and progress in this domain.
>
> We hope that the above point helps to clarify the extent and focus of the contribution of the proposed work.
>
> Now, to address the specific concerns of the reviewer:
>
> 1. Our work, at this point, is indeed purely experimental but this was intentional. On one hand, we wanted to establish first the utility of our general methodology in large-scale analysis of GAN distributions. On the other hand, we believe there is a need to expand first the experimental foundations of the topic before we are able  build a principled and well-grounded theory (as discussed above). In particular, we believe that our experimental insights and a precise way to capture some of the key aspects of GAN distribution diversity will exactly provide such foundations for the follow-up theoretical work.
>
> 2. We would like to clarify that we do not claim that the discovery of mode collapse is a contribution of our work. In particular, prior work exploring this phenomenon has been discussed in Section 1. As discussed above, our goal is to provide a methodology for capturing and quantifying this phenomenon in state-of-the-art GAN settings.
>
> 3. We compare our method to Inception scores for all diversity experiments (Tables 1 and 3). One should keep in mind, however, that Inception scores are computed based on labels from an Inception network pre-trained on ImageNet. It is not clear if these labels are meaningful for conventional GAN datasets like LSUN or CelebA, where images belong to the same high-level category (faces, rooms). Furthermore, a model could attain a high inception score even if it produces a single compelling image and does not capture the dataset diversity (Che et al. (2017), Hendrycks & Basart (2017)). Our methodology thus puts the GAN distributions through a more stringent test by analyzing how well they capture key characteristics of the underlying classes or modes in the true distribution. We will include a comparison to the metric proposed in Che et al. (2017) in the revised version of our paper. However, even the authors of that work suggest that their metric may not be appropriate for the CelebA and LSUN datasets (which are the datasets used in our paper). We would like to also emphasize that our choice of these datasets is deliberate, given that most GAN research is centered around these datasets, while most popular evaluation metrics seem better suited to other datasets such as CIFAR-10 and ImageNet.

---

### Decision · Program_Chairs · 2018-01-29
**ICLR 2018 Conference Acceptance Decision**

**Decision:**

Reject

**Comment:**

The paper proposes a new metric to measure GAN performance by training a classifier on the true labeled dataset and then comparing the distribution of the labels of the generated samples to the true label distribution. Reviewers find that the paper is well written but lacks novelty and is quite experimental does not present any new insights. The paper investigates well-known model collapse and diversity issues. Reviewers are not convinced that this is a good metric to measure sample quality or diversity as the generator can drop examples far away from the boundary and still achieve a good score on this metric.